# New Insight into Bone Immunity in Marrow Cavity and Cancellous Bone Microenvironments and Their Regulation

**DOI:** 10.3390/biomedicines13102426

**Published:** 2025-10-03

**Authors:** Hongxu Pu, Lanping Ding, Pinhui Jiang, Guanghao Li, Kai Wang, Jiawei Jiang, Xin Gan

**Affiliations:** 1Department of Orthopedics, Tongji Hospital, Tongji Medical College, Huazhong University of Science and Technology, Wuhan 430030, China; hongxu_pu@hust.edu.cn (H.P.); wangkai_hust@tjh.tjmu.edu.cn (K.W.); 2025tj0043@hust.edu.cn (X.G.); 2Orthopedic Hospital, The First Affiliated Hospital, Jiangxi Medical College, Nanchang University, Nanchang 330006, China; dinglanping1@jxutcm.edu.cn (L.D.); 4267124170@email.ncu.edu.cn (P.J.)

**Keywords:** osteoimmunity, marrow cavity, innate immunity, acquired immunity

## Abstract

Bone immunity represents a dynamic interface where skeletal homeostasis intersects with systemic immune regulation. We synthesize emerging paradigms by contrasting two functionally distinct microenvironments: the marrow cavity, a hematopoietic and immune cell reservoir, and cancellous bone, a metabolically active hub orchestrating osteoimmune interactions. The marrow cavity not only generates innate and adaptive immune cells but also preserves long-term immune memory through stromal-derived chemokines and survival factors, while cancellous bone regulates bone remodeling via macrophage-osteoclast crosstalk and cytokine gradients. Breakthroughs in lymphatic vasculature identification challenge traditional views, revealing cortical and lymphatic networks in cancellous bone that mediate immune surveillance and pathological processes such as cancer metastasis. Central to bone immunity is the neuro–immune–endocrine axis, where sympathetic and parasympathetic signaling bidirectionally modulate osteoclastogenesis and macrophage polarization. Gut microbiota-derived metabolites, including short-chain fatty acids and polyamines, reshape bone immunity through epigenetic and receptor-mediated pathways, bridging systemic metabolism with local immune responses. In disease contexts, dysregulated immune dynamics drive osteoporosis via RANKL/IL-17 hyperactivity and promote leukemic evasion through microenvironmental immunosuppression. We further propose the “brain–gut–bone axis” as a systemic regulatory framework, wherein vagus nerve-mediated gut signaling enhances osteogenic pathways, while leptin and adipokine circuits link marrow adiposity to inflammatory bone loss. These insights redefine bone as a multidimensional immunometabolic organ, integrating neural, endocrine, and microbial inputs to maintain homeostasis. By elucidating the mechanisms of immune-driven bone pathologies, this work highlights therapeutic opportunities through biomaterial-mediated immunomodulation and microbiota-targeted interventions, paving the way for next-generation treatments in osteoimmune disorders.

## 1. Introduction

Bone biology is a discipline that studies the structure, function, metabolism, development, repair, and related diseases of bone tissue. It encompasses the biological characteristics of bones and their relationship with systemic physiology [1]. Functionally, bones not only provide necessary support for the human body but also serve as the foundation for muscle movement and protect vital organs [2]. Additionally, bones have important immune functions, serving as critical sites for immune cell development, as well as playing significant roles in hematopoiesis and endocrine regulation. Bone immunity embodies a dynamic interplay between skeletal homeostasis and systemic immune regulation [3]. With the deepening of research on bone function, the concept of bone immunity continues to be updated [4].

Morphologically, bones are classified into long bones, short bones, flat bones, and irregular bones, each with distinct developmental processes. During embryonic development, all human bones originate from the mesoderm. For example, long bones like the femur and humerus, as well as short bones like carpals, develop through endochondral ossification, whereas flat bones such as cranial bones develop through intramembranous ossification. Irregular bones exhibit features of both ossification processes. Taking long bones as an example, their anatomical structure includes the diaphysis, epiphysis, and marrow cavity [5]. Traditionally, the marrow cavity was considered the primary site of bone immunity. However, with technological advances, immune cells have been discovered in the diaphysis and epiphysis. Lymphatic vessels in the cortical bone of the diaphysis, along with blood vessels, regulate bone formation and resorption, maintaining bone’s dynamic balance. The cancellous bone in the epiphysis are abundant in immune cells, contributing to immunoendocrine regulation [6]. Moreover, nerves densely distributed in bones regulate the immune environment within bones via neuroendocrine mechanisms [7]. These new findings reveal that the spatial immune environment of bones is not confined to the marrow cavity. The involvement of non-traditional immune cells in bone immunity has significantly expanded our understanding of the bone immune system (Figure 1).

This article summarizes the latest advances in bone immunology in recent years, focusing on the immune microenvironment within cancellous bone, the roles of various immune cells, immunoendocrine regulation, and the contributions of nerves, blood vessels, and lymphatic channels to bone immunity. It analyzes the synergistic regulatory interactions among these systems in maintaining bone homeostasis. Furthermore, the review explores the pathogenesis of several specific orthopedic diseases—such as osteoporosis, osteoarthritis (OA), and hematological malignancies—from an immunological perspective, aiming to provide insights for the development of novel therapeutic strategies.

## 2. Bone Marrow Cavity and Cancellous Bone Immunity

### 2.1. Immunity of the Bone Marrow Cavity

Bones are classical central immune organs, with the marrow cavity being a hollow space within the diaphysis that contains red and yellow marrow. This space is rich in hematopoietic cells, adipocytes, stromal cells, and a dense vascular network, primarily composed of sinusoidal capillaries. These elements facilitate hematopoiesis, immune cell generation and storage, and fat storage. The bone marrow serves as the origin of both innate and adaptive immune cells, including the development and maturation of B cells, T cells, granulocytes, macrophages, Natural Killer (NK) cells, and dendritic cells (DCs). Hematopoietic stem cells (HSCs) differentiate into common myeloid progenitors (CMPs) or common lymphoid progenitors (CLPs) under the influence of CXCL12 (SDF-1) and stem cell factor (SCF) secreted by bone marrow stromal cells [8].

CMPs differentiate into granulocyte-monocyte progenitors (GMPs) in response to GM-CSF and IL-3, activating signaling pathways such as JAK-STAT and MAPK/ERK, promoting nuclear translocation of transcription factors PU.1 and C/EBPα, and initiating the expression of CSF1R and CSF3R [9]. Under the influence of GM-CSF and G-CSF, GMPs generate neutrophil precursors that mature into neutrophils. These are released into the bloodstream to participate in pathogen phagocytosis and inflammatory responses. Eosinophils and basophils also derive from GMPs under the influence of IL-5 and IL-3, respectively, playing key roles in parasitic infection responses and inflammation regulation [10]. GMPs further differentiate into monocytes, which migrate to tissues and differentiate into macrophages (M1 or M2 type) under the action of M-CSF, participating in pathogen phagocytosis, antigen presentation, and immune regulation [11]. DCs can originate from GMPs or monocytic precursors, with FLT3L and IL-4 as critical factors. Mature DCs capture antigens and migrate to lymph nodes to initiate adaptive immune responses [12]. On the other hand, under the influence of IL-15 and IL-7, CLPs differentiate into natural killer cell precursors (NKPs), which subsequently mature into NK cells. NK cells undergo differentiation from CLPs to mature NK cells in the bone marrow, with IL-15 and Notch signaling being the core regulatory factors for their development. In innate immunity, NK cells directly kill virus-infected and tumor cells, as well as secrete cytokines to regulate adaptive immune responses, playing essential roles in antiviral, antitumor, and inflammation regulation [13]. The development of B cells is completed entirely within the bone marrow, where CLPs differentiate into pro-B cells under the influence of IL-7. Pro-B cells undergo immunoglobulin heavy chain gene rearrangement mediated by Rag1/2 enzymes to form pre-B cell receptors (pre-BCRs). They then proceed to the pre-B stage to complete light chain gene rearrangement, forming fully functional B cell receptors (BCRs). Immature B cells expressing IgM undergo negative selection to eliminate self-reactive B cells before migrating to peripheral organs (e.g., spleen) for maturation, ultimately expressing both IgM and IgD [14]. The processes are shown in the scheme below (Figure 2).

T cell development occurs only in the early stages within the bone marrow. CLPs differentiate into pro-T cells under the influence of IL-7 in the bone marrow and then migrate to the thymus, where TCR rearrangement and positive/negative selection occur, resulting in mature CD4^+^ or CD8^+^ T cells. The bone marrow microenvironment supports the development of B and T cells by secreting factors such as IL-7, SCF, and CXCL12, and providing vascular and stromal support, thus ensuring the functional integrity of adaptive immunity [15].

Recent research indicates that the bone marrow cavity not only serves as the site of immune cell generation but also acts as a reservoir for memory T and B cells [16,17]. This function depends on CXCL12 (SDF-1) secreted by bone marrow stromal cells, which attracts and retains memory T cells via the CXCR4 receptor. Stromal cells also secrete survival factors like IL-7 and IL-15, providing survival signals that maintain memory T cells in a dormant state. Upon re-encountering antigens, these dormant memory T cells can be rapidly reactivated and migrate to infection sites to participate in secondary immune responses. Similarly, memory B cells are stored in the bone marrow, where long-lived plasma cells secrete antibodies, providing sustained immunity for secondary responses [18]. These findings highlight the role of the bone marrow cavity in long-term immune memory.

Moreover, the bone marrow cavity is a major site for neutrophil storage and release. During infection or inflammation, sympathetic nerves regulate CXCR4 expression via norepinephrine signaling, promoting neutrophil release from the bone marrow cavity into the bloodstream, where they participate in pathogen defense [19]. Neutrophils released from the bone marrow not only clear inflammatory sites but also play crucial roles in tissue repair, such as promoting angiogenesis and bone injury healing. This reflects the neuroendocrine-immune regulatory functions of bones. Bone immunity also regulates HSC proliferation. Chronic inflammation stimulates bone marrow T cells to secrete IFN-γ and TNF-α, driving HSC differentiation into myeloid cells, potentially leading to myelodysplastic syndromes (MDS). Conversely, bone marrow macrophages maintain HSC quiescence through the secretion of CXCL12 and TGF-β, preventing overactivation and depletion of HSCs. The CXCL12-CXCR4 axis governs HSC migration and distribution [20].

The interplay between immune cells and HSCs in immune-hematopoietic balance is critical in chronic inflammation and tumor-associated bone marrow suppression. Additionally, as a hematopoietic organ, the bone marrow’s hematopoietic barrier isolates immune cells, enabling tumor cells to evade immune surveillance. Myeloid-derived suppressor cells (MDSCs) in the bone marrow suppress T and NK cell activity through IL-10 and TGF-β secretion, facilitating tumor cell colonization in the bone marrow [21]. Research suggests that inhibiting MDSC activity with CSF-1R inhibitors can activate NK cells in the bone marrow, effectively preventing tumor bone metastases [22]. Celia Dobersalske et al. reported that CD8+ T cells were abundant in the cranial bone marrow of glioblastoma patients, highlighting the bone marrow’s role in recruiting and activating immune cells in tumor immunity [23]. These studies reveal that the bone marrow cavity is not only a site for immune cell development but also a critical hub for immune responses.

### 2.2. Immunity in Cancellous Bone Microenvironment

Cancellous bone is predominantly found at the ends of long bones in the epiphysis. It consists of a network of interconnected trabeculae forming a honeycomb-like structure, interspersed with nerves and blood vessels. Anatomically, cancellous bone and the marrow cavity are directly connected, enabling functional collaboration through shared vascular and bone endosteum networks [24,25]. The trabeculae are covered by the endosteum, which contains osteoblasts, osteoclasts, and mesenchymal stem cells [26]. The endosteum extends into the marrow cavity, forming a structural continuity that allows for the coordinated exchange of cytokines, nutrients, and signaling molecules [27]. Despite their structural connection, cancellous bone and the marrow cavity differ significantly in function. Cancellous bone, composed of trabeculae with lower Young’s modulus (2–5 GPa) compared to cortical bone, provides mechanical support while serving as the primary site for bone remodeling. Osteoblasts and osteoclasts in cancellous bone maintain the balance of bone resorption and formation, releasing matrix-stored biofactors (e.g., TGF-β, IGF-1) that regulate immune cell functions [28].

The role of immune cells in regulating bone resorption and formation has been extensively studied. Granulocytes secrete cytokines such as IL-17 and TNF-α to directly modulate osteoclast activity, thereby enhancing bone resorption [29]. Moreover, granulocytes facilitate pathogen clearance through chemokine secretion and rapid release of IL-8 and ROS [30]. T cells, particularly Th17 cells, promote osteoclast differentiation by secreting IL-17, which upregulates RANKL expression and accelerates bone resorption [31,32,33]. In contrast, regulatory T cells (Tregs) secrete IL-10 and TGF-β to suppress excessive immune responses, promoting bone formation and repair [34]. B cells, besides their key roles in immune responses, also secrete antibodies and RANKL to activate osteoclasts directly [35,36].

Macrophages play a pivotal role in cancellous bone, particularly at the trabecular surfaces near osteoblasts and osteoclasts-regions critical for bone remodeling and metabolism [37,38]. Macrophages polarize into different subtypes (M1 or M2) depending on microenvironmental signals. M1 macrophages, activated by Th1-derived IFN-γ or bacterial LPS, secrete pro-inflammatory cytokines (e.g., TNF-α, IL-1β, IL-6) and nitric oxide (NO), enhancing bone resorption. Conversely, M2 macrophages, polarized by signals from Th2 cells, Tregs, or fibroblasts, secrete anti-inflammatory factors like IL-10 and TGF-β, promoting immune suppression, tissue repair, and inflammation resolution [39,40,41,42].

Macrophages can also differentiate into osteoclasts under the influence of RANKL and M-CSF, contributing to bone resorption. Immune cells interact via cytokine release to regulate bone metabolic balance. Pro-inflammatory cytokines like TNF-α, IL-1β, IL-6, and IL-17 promote bone resorption and inhibit osteoblast activity, whereas anti-inflammatory cytokines like TGF-β and IL-10 suppress osteoclast function and facilitate bone repair. RANKL, a critical osteoclastogenic factor, binds to its receptor RANK to activate osteoclast differentiation and bone resorption [43]. VEGF, a key angiogenic factor, supports bone remodeling by promoting vascularization [44]. These cytokines and cellular interactions within cancellous bone form a complex immune microenvironment that maintains bone metabolic balance (Figure 3).

Cancellous bone is one of the most immunologically active areas within bone tissue. Ning Yang et al. have comprehensively reviewed the role of bone immunity in bone regeneration, highlighting how emerging technologies have enriched the concept of bone immunity [4]. With advances in single-cell and spatial transcriptomics, the challenges of dissociating bone tissue have been overcome, and high-resolution sequencing technologies have revealed the immune functions of cancellous bone. However, due to the complete degradation of nuclei and cytoplasm after calcium deposition in bone cells, these cells are undetectable in single-cell atlases. Qin Ling et al. used scRNA and CODEX analysis on femoral head cells, identifying significant heterogeneity in composition. Macrophages accounted for 10–20%, granulocytes for 15–25%, dendritic cells for 2–5%, and NK cells for 1–3%. Among adaptive immune cells, T cells constituted 20–30%, and B cells 5–10%. Additionally, myeloid-derived suppressor cells (MDSCs) accounted for 1–3%, significantly increasing during tumors or chronic inflammation [45].

The immune microenvironment of cancellous bone regulates bone homeostasis and dynamic metabolic balance. Cells in cancellous bone are not randomly distributed but exhibit specific spatial aggregation patterns. For example, macrophages concentrate on trabecular surfaces, aligning with their roles in bone metabolism. T and B cells predominantly cluster near blood vessels, while endothelial cells are closely associated with lymphocytes and myeloid cells. This complex network of immune cells and cytokines constitutes the cancellous bone immune microenvironment.

### 2.3. Differences in the Spatial Microenvironment Between Cancellous Bone and Bone Marrow Cavity

Based on the spatial transcriptomic analysis of murine bone marrow by Xue Xiao et al., significant distinctions between the bone marrow cavity and cancellous bone regions were elucidated. The trabecular bone compartment is characterized by the enrichment of mesenchymal lineage cells, such as osteoblasts and osteoprogenitors, which highly express bone formation-related genes including Col1a1, Bglap, Sp7, and Runx2, indicating active osteogenesis and matrix remodeling. In contrast, the bone marrow cavity predominantly harbors hematopoietic cells and CXCL12-abundant reticular (CAR) cells, with upregulated expression of hematopoietic markers like Ptprcand Hba-a2, reflecting its primary role in supporting hematopoiesis and immune cell production. Signaling pathway analysis further revealed that BMP/TGFβ pathways are more active near trabecular bone surfaces, promoting osteogenic differentiation, while WNT/PDGF signaling dominates in the marrow cavity, facilitating angiogenic and niche functions. Metabolic profiling showed a preference for oxidative phosphorylation and fatty acid metabolism in trabecular regions, whereas glycolytic processes are heightened in the marrow cavity to meet the energetic demands of proliferating hematopoietic cells. These spatial heterogeneities underscore the functional specialization within bone microenvironments, where trabecular bone serves as a “bone-forming center” and the marrow cavity acts as a “hematopoietic hub”. This compartmentalization highlights the importance of spatial context in regulating stem cell fate and tissue homeostasis [46].

Takahito Iga et al. identified spatial heterogeneity of endothelial cells through single-cell sequencing and discovered a capillary subtype exclusively present in the epiphyseal cancellous bone, termed type S (secondary ossification) endothelial cells (ECs). Type S ECs possess unique phenotypic characteristics in terms of structure, plasticity, and gene expression profiles. Genetic experiments demonstrated that type S ECs atypically contribute to the acquisition of bone strength by secreting type I collagen, the most abundant component of the bone matrix. Moreover, these cells form a distinct reservoir for hematopoietic stem cells [47].

Julia Tilburg et al. performed spatial transcriptomic analysis and found that megakaryocytes in the proximal and distal cancellous bone of murine femurs highly express Ppbp, Pf4, Tmsb4xand Rps11, Phgdh, respectively [48]. These studies collectively indicate significant heterogeneity in cellular composition and function between cancellous bone and the bone marrow cavity.

## 3. Regulation of Bone Immune Homeostasis

### 3.1. Brain-Gut-Bone Regulation and Mechanisms in Bone Immunity

The relationship between the brain and bones has been widely reported in clinical settings. For example, patients with multiple injuries involving bone fractures and brain trauma experience faster fracture healing. Studies by Xiong Yuan and Bai Xiaochun revealed that the injured brain releases exosomes containing miR-21-5p, miR-328a-3p, and miR-150-5p, which target the 3′UTR regions of SMAD7, FOXO4, or CBL genes, thereby accelerating bone regeneration [49,50]. Cao Xu et al. proposed the concept of the “brain–bone axis,” a regulatory network connecting the brain, dorsal root ganglia (DRG), and skeletal system. This axis emphasizes the brain’s regulation of bone metabolism through neural transmission, immune responses, and neurotransmitters, highlighting the reciprocal feedback between the nervous and skeletal systems [51]. The DRG, as a sensory nerve hub, mediates the perception of bone injuries, immune responses, and bone repair, transmitting nerve impulses to the brain, which regulates bone metabolism via sympathetic and parasympathetic nerves. The discovery of the brain–bone axis has opened new avenues for cancellous bone immunity research [52,53]. The balance of osteoclasts and osteoblasts, the polarization effect of macrophages, and the activation of T cells are the main effector targets of these regulatory effects.

The nervous system influences bone immunity primarily through neurotransmitters and neuropeptides. The sympathetic nervous system regulates immune responses by releasing norepinephrine (NE). Beta-adrenergic blockers enhance the anabolic effects of parathyroid hormone (PTH), promoting fracture recovery [54]. NE stimulates immune cells by binding to β-adrenergic receptors, upregulating RANKL secretion, and promoting osteoclast differentiation, leading to increased bone resorption. β-adrenergic receptors, particularly β2-AR and β1-AR, modulate immune cell activity, while β3-AR promotes HSC proliferation. Zhang Yingze’s research identified that β2-AR mediates M2 macrophage polarization via Adrb2 and Adrb3, with αCGRP also playing a significant role in this process, as reported by Denise Jahn [55,56]. Immune cells widely express β-adrenergic receptors, which are G protein-coupled receptors (GPCRs) that activate the cAMP signaling pathway, enhancing macrophage activation, antigen presentation by DCs, and T cell activation [57,58]. In addition, the adrenergic–CXCL12 axis is also the most classic pathway for sympathetic regulation of hematopoiesis [59]. Sympathetic nerve endings in the bone marrow release the neurotransmitter norepinephrine in a circadian rhythm, and its concentration reaches its peak during the active period of the body. Norepinephrine reduces the expression of CXCL12 by acting on Adrb3 on stromal cells in the hematopoietic microenvironment. This enables the HSC to be anchored in the microenvironment [60]. The periodic down-regulation of CXCL12 weakens the restraint on HSCS, allowing HSCS and mature immune cells such as neutrophils to be released from the bone marrow into the peripheral circulation at regular intervals every day according to the circadian rhythm to perform immune surveillance functions [61,62]. Besides, the sympathetic nerve endings in the bone marrow can also release dopamine. Unlike norepinephrine, which mainly acts on niche cells, dopamine directly acts on HSCS and progenitor cells (HSPCs), exerting its influence through dopamine receptor 2/3 on their surfaces [63]. Knocking out these receptors will lead to the depletion of the HSPC pool, which indicates that continuous dopaminergic signaling is crucial for maintaining the integrity and quantity of the stem cell bank [64]. This function exists in parallel with the adrenergic pathway that regulates cell outflow, jointly ensuring the homeostasis of the hematopoietic system.

The parasympathetic nervous system releases acetylcholine (Ach), which modulates immune cell function. Ach binds to M1-type cholinergic receptors (α7nAChR) on immune cells, suppressing M1 macrophage polarization and reducing the secretion of inflammatory cytokines such as TNF-α, IL-1β, and IL-6. Ach also decreases IL-17 secretion by Th17 cells, thereby dampening immune responses [65,66,67]. For hematopoietic function, the regulation of daily migration of HSCS by cholinergic signaling exhibits a more complex biphasic pattern, involving two different signaling sources and mechanisms of action [68]. Parasympathetic cholinergic signals from the central nervous system dominate at night. It inhibits the norepinephrine tension of the sympathetic nerve in the bone marrow through systemic action, thereby restricting the outflow of HSCS and promoting their retention and homing in the niche. During the day, under the trigger of light, the sympathetic cholinergic fibers in the local bone marrow are activated. The ACh it releases collaborates to support the HSC outflow process driven by norepinephrine by inhibiting the expression of vascular cell adhesion molecules in the hematopoietic microenvironment [60,69].

The sensory nervous system also plays an important role in efferent regulation in the bone marrow, especially during the forced mobilization of the HSC. G-CSF is a key drug used in clinical practice to mobilize HSCS for transplantation, and its mechanism of action was once considered to mainly depend on SNS. However, recent studies have clearly pointed out that nociceptive (pain) nerves are essential for G-CSF-induced HSC mobilization. When the body is stimulated by G-CSF, the nociceptive nerves in the bone marrow are activated and release the neuropeptide -CGRP. This mechanism is fundamentally different from the SNS channel. Norepinephrine regulates HSC by indirectly acting on niche cells, while CGRP directly acts on HSC itself. Research has found that the surface of HSC expresses a receptor complex of CGRP, which is composed of receptor activity-modified protein-1 (RAMP1) and calcitonin receptor-like receptor (CALCRL). After CGRP binds to this receptor, it activates the Gαs/ adenylate cyclase /cAMP signaling pathway within HSC. This signal directly promotes HSC to break away from the ecological niche and enter the peripheral blood circulation.

Besides the direct regulation of the nervous system, neuropeptides also play an important role in the regulation of bone immunity. Substance P (SP), a neuropeptide secreted by sensory nerves, regulates bone and cartilage metabolism, as well as fracture healing [70]. SP is recognized by both the nervous and immune systems, with bone and cartilage cells capable of synthesizing and secreting this neuropeptide. It acts through autocrine or paracrine mechanisms to promote target cell proliferation, differentiation, apoptosis, matrix synthesis, and degradation [70,71]. Neuropeptide Y (NPY), secreted by sympathetic nerves, exhibits anti-inflammatory effects and plays key roles in the regulation of the nervous, immune, and physiological systems [72]. As a neurotransmitter, NPY participates in appetite control, energy balance, stress responses, and immune regulation. By binding to receptor subtypes such as Y1, Y2, and Y5, NPY inhibits immune cell activation, reducing bone resorption and protecting bone homeostasis [73,74]. NPY also plays a significant supporting role in the hematopoietic function of the bone marrow cavity microenvironment, directly regulating the recruitment of HSC and bone marrow reconstruction.

Enkephalins, a class of endogenous opioids secreted by the central nervous system, have analgesic properties. By binding to μ-opioid receptors, they modulate sympathetic and parasympathetic functions, suppress inflammation, and regulate bone resorption [75]. These findings illustrate the intricate mechanisms underlying bone-brain interactions.

The interaction between gut microbiota and bone metabolism has been widely documented. Increasing evidence suggests that gut microbiota not only affects intestinal health but also exerts significant influence on the skeletal system by modulating immune responses, endocrine signaling, and metabolic pathways [76]. Advances in metabolomics and microbiomics have provided new methods for exploring the functions of gut microbiota. Studies have demonstrated that gut microbiota is closely associated with bone immunity, bone metabolism, and bone remodeling through various mechanisms [77]. Moreover, the gut possesses an autonomous nervous system, often referred to as the “second brain.” The concept of the “brain–gut–bone axis” has been proposed, where the vagus nerve and autonomic nervous system (ANS) serve as the bidirectional communication core between the brain and gut. The vagus nerve senses changes in gut microbiota and transmits signals to the brain while regulating gut motility, secretion, and barrier function, thus influencing the gut environment [78]. The enteric nervous system (ENS) communicates with the central nervous system (CNS) by regulating gut motility and secretion. ENS neurons directly sense gut microbiota-derived metabolites (e.g., short-chain fatty acids) and influence neurotransmitter release (e.g., serotonin), affecting brain function and behavior [79]. Furthermore, the brain regulates gut barrier and microbiota composition through neuroendocrine mechanisms, while gut metabolites feedback to modulate brain function, creating a bidirectional relationship between the gut and bone [80,81].

Gut microbial metabolites such as short-chain fatty acids (SCFAs)—including butyrate, propionate, and acetate—are products of microbial fermentation of dietary fiber [82]. Butyrate and propionate act as histone deacetylase (HDAC) inhibitors, reducing Th17-derived IL-17 secretion and promoting Treg proliferation by inhibiting sequences like Foxp3 [83,84]. SCFAs also inhibit M1 macrophage polarization, promote M2 polarization, decrease the secretion of TNF-α and IL-6, and enhance IL-10 production [85]. Indole and its derivatives, which are tryptophan metabolites produced by gut microbiota, activate the aryl hydrocarbon receptor (AhR). Upon activation, AhR translocates to the nucleus, forms a heterodimer with aryl hydrocarbon receptor nuclear translocator (ARNT), and binds to AhR response elements (AhREs) in DNA, regulating the expression of cytochrome P450 family CYP1 genes [85,86]. AhR activation promotes Treg differentiation, inhibits Th17 activity, balances Th1/Th2 responses, and regulates NF-κB signaling to foster M2 macrophage polarization while reducing M1 activity. This process decreases osteoclastic bone resorption [87].

Polyamines such as putrescine, spermidine, and spermine are small molecules produced through the metabolism of amino acids by gut microbiota. These polyamines influence bone tissue through systemic circulation [88]. Spermidine, for instance, upregulates Foxp3 expression, promoting Treg proliferation and differentiation while reducing inflammation in the bone microenvironment. Polyamines also decrease Th17-derived IL-17 secretion and shift macrophage polarization from M1 to M2, reducing TNF-α and IL-6 levels. These mechanisms are particularly relevant in the pathogenesis of OA [89]. Furthermore, polyamines suppress RANKL signaling, decreasing osteoclast activity and inhibiting bone resorption, making them potential therapeutic molecules for restoring bone metabolic balance and preventing osteoporosis and inflammatory bone diseases. [88,89].

The mucosa-associated lymphoid tissue (MALT) in the digestive tract is the largest mucosal immune system in the body. It is directly connected to the external environment and plays key roles in barrier protection, immune regulation, and interaction with microbiota. MALT-derived B1 cells produce sIgA, which indirectly modulates RANKL expression by inhibiting pro-inflammatory factors, thereby reducing osteoclast activation [90]. Additionally, the gut has an immunoendocrine function, secreting factors such as IGF-1 to regulate bone immunity [91]. The gut and bone form a coordinated feedback network, and the “brain–gut–bone axis” has gradually become a recognized concept. Given the complexity of the gut microbiota, which comprises over 2000 species, coupled with the pleiotropic nature of most biologically active molecules produced or induced by them, the interaction between the gut microbiota and bone requires further exploration.

Immunoendocrine regulation is an integral component of bone immunity. The endocrine system regulates metabolism throughout the body, with endocrine factors playing key roles in balancing bone metabolism and immune responses through complex mechanisms. The regulatory effects of hormones such as estrogen, parathyroid hormone (PTH), thyroid hormones, and vitamin D on bone immunity have long been recognized [92]. Estrogen, for instance, binds to ERα and ERβ receptors on osteoclast membranes, reducing osteoclast activity. It also suppresses the generation of pro-inflammatory Th17 cells while promoting the proliferation of regulatory T cells (Tregs) [93]. Testosterone, a crucial sex hormone, works synergistically with estrogen during puberty to promote bone mass accumulation. testosterone stimulates the development of osteoblasts via the enhancement of IL-1β signaling in precursor cells [94,95]. Growth hormone exerts its critical effects on both skeletal growth and adult bone homeostasis largely through the liver-derived insulin-like growth factor-1 (IGF-1), in a process known as the GH/IGF-1 axis [94].

Clinically, hyperthyroidism is strongly associated with increased osteoporosis risk. Thyroid hormones (T3/T4) promote osteoclast formation via thyroid hormone receptors (TRs). Excessive thyroid hormones accelerate bone resorption, leading to decreased bone density, with MCT8 and MCT10 playing key roles in hormone transport within bone tissue [96]. Additionally, thyroid hormones regulate Th1 cell activity, indirectly influencing bone immunity [97]. Parathyroid glands, located on the lateral sides of the thyroid gland, secrete PTH, which primarily regulates bone calcium and phosphate metabolism. PTH has complex effects on bone immunity: short-term elevation of PTH enhances osteoblast activity and bone formation, whereas chronic high levels increase RANKL expression, promoting osteoclastogenesis and bone resorption. In the immune system, PTH stimulates T cell proliferation, particularly Th1 and Th17 cells [98]. Elevated PTH levels directly or indirectly amplify bone resorption by enhancing immune cell activity, thereby influencing the bone immune microenvironment [99]. Recent studies suggest that gut microbiota-produced SCFAs, especially butyrate, play a crucial role in PTH-induced osteoblast proliferation and bone formation [100]. These findings provide insights into the mechanisms underlying PTH therapy for osteoporosis and highlight the broader role of the gut microbiota in immune regulation [101]. Calcitonin, secreted by the parafollicular cells (C cells) of the thyroid gland, directly inhibits the activity and differentiation of osteoclasts. This action reduces bone resorption and promotes the deposition of calcium in bones [94].

Cortisol, as a stress hormone, exerts its effects on bone immunity primarily through immune suppression and increased bone resorption. Chronic high cortisol levels inhibit osteoblast function, enhance osteoclast activity, and cause bone loss [102]. Simultaneously, cortisol influences immune cell functions, exacerbating inflammatory responses and further driving bone resorption [103,104]. The intricate interplay of endocrine factors forms a dense immunoendocrine network, dynamically regulating bone immunity to maintain bone metabolic balance.

Bone marrow is rich in white adipocytes, constituting approximately 30% of marrow cells. These cells not only serve as energy reservoirs but also secrete adipokines to regulate bone immunity and hematopoiesis. They influence immune responses by releasing cytokines such as IL-6 and leptin [105]. Leptin is a pleiotropic hormone that serves as a core hub connecting the metabolic, immune, and skeletal systems [106]. Peripherally, it acts directly on bone cells to exert an anabolic effect that promotes bone formation, through mechanisms such as inhibiting RANKL expression [107]. Centrally, it exerts a catabolic effect by inhibiting bone formation and promoting bone resorption through the hypothalamic-sympathetic nervous pathway, which facilitates RANKL expression [108]. The ultimate skeletal effect of leptin is the result of a balance between these two opposing pathways. Furthermore, as a pro-inflammatory cytokine, leptin can activate innate and adaptive immune responses, particularly by driving the differentiation of T cells towards Th1 and Th17 lineages. These T cell subsets are potent sources of RANKL, thus forming a direct pathway that links adipose tissue, the immune system, and bone resorption [109]. leptin also activates downstream signaling pathways such as JAK-STAT, PI3K-AKT, MAPK/ERK, AMPK, and NF-κB, promoting B cell proliferation and antibody secretion. It also enhances Th1 and Th17 cell differentiation, increasing the secretion of cytokines like IFN-γ and IL-17, which ultimately boost osteoclast activity and bone resorption [110]. Single-cell sequencing studies by Kara and Gao have identified a population of LepR(+) stromal cells in the bone marrow that secrete NGF in response to leptin, promoting bone marrow regeneration [111,112]. Bone marrow mesenchymal stem cells exhibit trilineage differentiation potential, differentiating into adipocytes under the influence of insulin and IGF, mediated by transcription factors PPARγ and C/EBPα [113]. In obesity, an increase in marrow adiposity induces a pro-inflammatory microenvironment, enhancing myeloid differentiation of HSCs, suppressing osteoblast activity, and contributing to osteoporosis [114].

The primary bone immunomodulatory factors mentioned above are summarized in Table 1.

### 3.2. The Tubular System and Bone Immunity

Blood vessels and lymphatic vessels are critical components of the circulatory system, providing nutrient delivery, waste removal, fluid regulation, and serving as key sites for immune activity. Blood and lymphatic vessels are present in most tissues, with exceptions like the cornea and cartilage. The cornea and lens, for example, rely on aqueous humor for nourishment and are devoid of vasculature and lymphatics [115]. Cartilage, being avascular, depends on synovial fluid and subchondral bone for nutrients. Zhang Feng’s research revealed that chondrocytes synthesize hemoglobin storage bodies (HEDYs) under hypoxic conditions, allowing them to survive in avascular environments without relying on conventional HIF pathways [116]. Similarly, the central nervous system (CNS) lacks traditional lymphatic vessels due to the blood-brain barrier. However, functional lymphatic vessels have been identified in the meninges (especially dura mater), aligning with venous sinuses and cranial nerve pathways. These vessels connect to cervical lymph nodes and play roles in cerebrospinal fluid clearance and immune regulation [117].

Bone tissue is richly vascularized, receiving over 10% of cardiac output. Capillaries serve as the primary sites of microcirculation in cancellous and marrow cavities. The metaphysis at the ends of long bones contains a dense vascular network, nourishing cancellous bone through small foramina. In cortical bone, Haversian and Volkmann canals form a dense microvascular network, supplying nutrients to cortical regions [25]. Bone marrow sinusoidal vessels create a hematopoietic microenvironment essential for immune cell generation, while vascular networks facilitate immune cell migration into peripheral tissues. Immune cell trafficking and H-type vessel formation, regulated by VEGF and Notch pathways, are critical for maintaining bone immune balance [118,119].

Historically, bones were thought to lack lymphatic tissue. This perception persisted until the early 21st century when advancements revealed the presence of lymphatic vessels in bone. One key finding was the role of lymphatic abnormalities in Gorham-Stout disease (GSD), characterized by intrabony lymphangiomas. GSD, also known as disappearing bone disease or massive osteolysis, is a rare idiopathic condition linked to abnormal lymphangiogenesis and local immune dysregulation, indirectly suggesting the presence of lymphatic vessels in bones [120,121].

The discovery of lymphatic vessels in bone was delayed due to technical challenges. Bone tissue’s calcification hindered the identification of single-layer lymphatic endothelial cells. Innovations such as tissue clearing and light-sheet microscopy eventually overcame these obstacles, enabling whole-organ imaging. Lincoln Biswas and colleagues identified continuous lymphatic channels within cortical bone and more abundant lymphatic pathways in cancellous bone [122,123,124]. These findings filled a significant gap in our understanding of bone immunity.

Lymphatic endothelial proliferation depends on the VEGFR3 signaling pathway, with LYVE-1 and PROX1 serving as markers of lymphatic endothelial cells. Lymphatic vessels are critical for antigen-presenting cell (APC) and lymphocyte trafficking, as well as cancer metastasis. The identification of lymphatic vessels in bone also clarified mechanisms underlying cancer metastasis to bone [125,126]. However, research on bone lymphatics remains sparse. Evidence suggests that enhanced lymphatic drainage can significantly promote fracture healing [127]. Despite these advancements, the existence of lymphatic vessels within the bone marrow cavity remains unconfirmed. The hematopoietic barrier in the marrow creates a distinct microenvironment isolated from external immune influences, which theoretically justifies the absence of lymphatic vessels in this region (Figure 4).

## 4. Discussion and Prospects

Bone immunity is critical for maintaining bone balance and homeostasis. Dysregulation of bone immunity exacerbates osteoporosis and leads to various diseases. Bone immunity plays a central role in the pathogenesis of numerous conditions, such as OA, osteoporosis, acute myeloid leukemia (AML), and aplastic anemia. Immune processes are central to nearly all disease mechanisms, with bone immunity intersecting with bone metabolism, regeneration, hematopoiesis, and cancer. Understanding bone immunity may uncover novel therapeutic strategies for various orthopedic diseases.

Fracture healing is a highly complex process consisting of the inflammatory, reparative and remodeling phases, which is precisely regulated by the immune system [128]. Imbalances in the immune microenvironment, such as chronic inflammation or immunosuppression, can lead to delayed or non-union of fractures. It is therefore crucial to understand the interplay between immune cells and osteocytes. Innate immune cells (e.g., neutrophils and macrophages) infiltrate rapidly after injury, with M1-type macrophages mediating the initiation of inflammation in the early phase, whereas M2-type macrophages promote tissue repair in the mid and late phases [129]. Adaptive immune cells (e.g., T cells and B cells) regulate the process of bone repair by secreting cytokines that influence osteoclast and osteoblast activity. In addition, regulatory T cells (Tregs) and Th2 cells promote osteogenesis by secreting anti-inflammatory factors such as IL-10 and TGF-β, whereas hyperactivation of Th1 cells can inhibit bone formation by promoting IFN-γ secretion [130]. Current strategies to improve bone repair focus on modulating the immune microenvironment using immunomodulatory biomaterials, including promoting M1 to M2 macrophage polarization, modulating the Th1/Th2 balance, and creating ecological niches conducive to bone regeneration [131,132,133].

In osteoporosis, bone metabolic imbalance is a primary cause, characterized by reduced bone mass and density. Immune cells, particularly macrophages, are pivotal in its pathogenesis. The balance between osteoblasts and osteoclasts is disrupted, with increased RANKL expression and decreased osteoprotegerin (OPG) levels enhancing osteoclast activity. Chronic inflammatory environments in the bone marrow drive macrophage polarization toward the M1 phenotype, increasing pro-inflammatory cytokines such as TNF-α, IL-1β, and IL-6. Under excessive RANKL stimulation, macrophages differentiate into osteoclasts, further intensifying bone resorption. Estrogen deficiency after menopause exacerbates these effects by activating Th1 and Th17 cells, which secrete RANKL and pro-inflammatory cytokines, amplifying osteoclast activity while suppressing osteoblast function [134]. The interaction between adipocytes and macrophages plays a crucial role in osteoporosis. Bone marrow adipocyte metabolism is a key factor in the pathogenesis of osteoporosis. Studies have shown that bone fat content in osteoporosis patients can increase by 30–70% compared to healthy individuals. Systemic lipid metabolism disruptions can lead to bone metabolic imbalances. Lu Ke discovered that lecithin-cholesterol acyltransferase (LCAT) in the liver–bone axis regulates lipid circulation, with elevated osteoclast activity observed in cirrhotic patients, thereby affecting bone metabolism [135]. Increased lipid metabolism not only inhibits the osteogenic differentiation of mesenchymal stem cells but also enhances osteoclast function through the secretion of factors such as leptin and adiponectin. Hui Peng found that mechanical stimuli induced by exercise promote the release of reticulocalbin-2 (RCN2) from macrophages, thereby stimulating bone marrow adipocyte metabolism and alleviating osteoporosis [136].

Clinical analyses have shown that patients with asthma and chronic pulmonary disease have higher rates of osteoporosis [137,138,139]. Recent research confirms this association, indicating that hypoxic microenvironments increase reactive oxygen species (ROS) accumulation, which promotes Th1 and Th17 cell activity through the HIF-1α signaling pathway. This process polarizes macrophages toward the M1 phenotype, thereby enhancing osteoclast differentiation, as well as ROS directly activate osteoclasts [140,141].

In summary, osteoporosis arises from the combined effects of metabolic imbalance and immune dysregulation. The immune system, through abnormal activation of osteoclasts by RANKL and pro-inflammatory cytokines (e.g., TNF-α, IL-17, IL-6), simultaneously suppresses osteoblast function, leading to excessive bone resorption and reduced bone formation. Immune regulation abnormalities, particularly the overactivation of T cells and Th17 cells, are key drivers of osteoporosis. Recent studies have found that whole-brain c-FOS fluorescence quantification can reflect osteoporosis severity [142]. These findings illustrate the intricate pathological interplay between bone and the immune system.

Local inflammation in joints is closely tied to bone immunity. Joint inflammation, often caused by infection, trauma, immune abnormalities, or metabolic disorders, results in various forms of arthritis, including rheumatoid arthritis (RA), OA, gouty arthritis, and infectious arthritis [143]. OA is characterized by non-specific inflammation involving cartilage degradation, synovial inflammation, and subchondral bone remodeling. Recent studies reveal distinct subpopulations of chondrocytes, with significant inflammatory differences among these groups. The transition from newborn cartilage to fibrous cartilage follows distinct sequential and stage-specific pathways [144].

Macrophages play a critical role in OA pathogenesis by driving bone resorption, remodeling, and inflammation. Synovial macrophages, particularly those polarized to the M1 phenotype, contribute significantly to OA progression. Recent findings show that fibroblasts in the synovium and fat pads secrete Midkine and APOE, regulating cartilage and subchondral bone function, thereby influencing OA development [145,146]. The subchondral bone microenvironment also plays a vital role in OA. Subchondral bone lies beneath the articular cartilage and consists of a bony layer (subchondral bone plate) and underlying cancellous bone. It supports cartilage while serving as a nutrient source. Immune cells in subchondral bone contribute significantly to OA pathogenesis.

Subchondral bone remodeling in OA is linked to excessive inflammation. M1 macrophages secrete IL-1β and IL-6, driving abnormal osteoblast activation. Activation of the Wnt/β-catenin pathway leads to osteophyte formation and subchondral bone sclerosis, with Th17 cells playing a central role in this process [147]. Recognizing OA as a systemic inflammatory disease emphasizes the central role of immune dysregulation in its pathology [148]. The pathogenesis of osteoarthritis was presented in Figure 5.

Most parts of the body exhibit immune cell infiltration and lymphatic vessel distribution. However, certain regions are exceptions, such as the brain, lens, testis (due to the blood-testis barrier), and ovarian follicles. Similarly, the bone marrow cavity creates an immune-privileged environment through its hematopoietic microenvironment, allowing immune cell development while shielding it from external immune influences [45]. This immune privilege is crucial for bone marrow functionality but also facilitates tumor survival and immune evasion.

Acute myeloid leukemia (AML) exemplifies this phenomenon. AML pathogenesis involves not only the proliferation and differentiation defects of leukemia stem cells (LSCs) but also modifications to the bone marrow microenvironment that support leukemic growth while evading immune surveillance. AML cells induce stromal cells to secrete CXCL12 and TGF-β, attracting Tregs and MDSCs, suppressing T and NK cell activity, and remodeling the marrow stroma via matrix metalloproteinases (MMPs) [149]. AML also induces abnormal angiogenesis through VEGF secretion, restricting immune cell infiltration and promoting endothelial cells to release IL-6 and TGF-β, further suppressing immune responses [150]. Additionally, AML upregulates CXCL12 secretion to shelter LSCs within marrow niches and suppress dendritic cell antigen presentation, reducing T cell activation [151].

When this immune privilege is disrupted, it can lead to autoimmune diseases such as multiple sclerosis, infertility, or sympathetic ophthalmia. Aplastic anemia (AA), characterized by immune-mediated bone marrow failure, exemplifies how immune imbalance can disrupt hematopoiesis. The primary pathology involves destruction of the hematopoietic microenvironment and exhaustion of HSCs. In AA, immune dysregulation fosters an inflammatory, immunosuppressive environment, leading to HSC damage, apoptosis, and impaired hematopoiesis [152,153,154].

Immunity is intricately linked to every aspect of bone biology. While significant progress has been made in understanding the development of immune cells within bone, many critical biological processes remain unexplored. These include the precise regulation of cellular trajectories for immune cell lineage selection, the mechanisms by which the bone marrow microenvironment maintains immune privilege, and the interactions between various cellular subpopulations within bone niches. The strong autofluorescence of bone tissue poses significant challenges for molecular biology techniques in bone research. Many such studies require decalcification before staining and sectioning. However, prolonged decalcification can impair immunogenicity, making these investigations particularly difficult. As a result, despite the advent of single-cell omics technologies over a decade ago, their application to bone biology has only recently emerged [155,156]. New discoveries rely on advancements in technology. In recent years, the resolution and functionality of omics tools have continuously improved, leading to breakthroughs in bone immunity research. Among these, single-cell resolution 3D spatial omics stands out as a promising tool. Its application in bone biology is highly anticipated and has the potential to deliver groundbreaking insights into bone immunity [157].

## 5. Conclusions

Immunology is a vast field, and bone immunity represents a specialized intersection between immunology and bone biology. It plays a vital role in bone-related diseases, exhibiting unique characteristics compared to other immune systems. Bones serve as sites for both immune cell generation and immune responses, with spatial and temporal integration of immune and metabolic processes. Bone immunity is intricately linked to metabolism, endocrine functions, and the nervous system, complicating its study. This review focused on the narrower definition of bone immunity involving immune cells and cytokines, summarizing key advances in bone marrow immunity, cancellous bone immunity, and related systemic diseases, such as osteoporosis, osteoarthritis, and hematological malignancies.

The discovery of lymphatic vessels within bone is a landmark advancement, though the existence of lymphatics within the bone marrow cavity remains unresolved. A deeper understanding of bone immunity could provide critical insights into novel therapeutic strategies for treating orthopedic and systemic diseases.

## Figures and Tables

**Figure 1 biomedicines-13-02426-f001:**
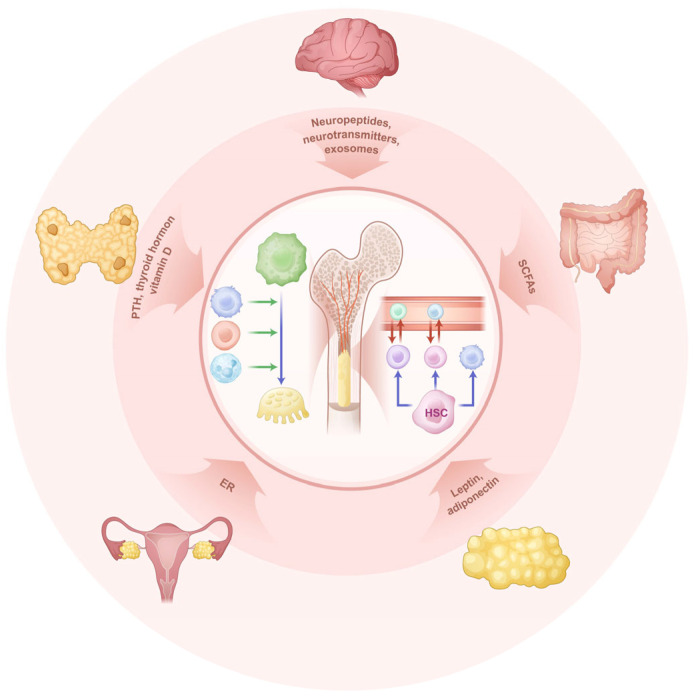
The scheme of the regulation of bone immunity. The outer ring indicates the organ systems that interact with bone immunity, including the parathyroid glands, ovaries, hypothalamic-pituitary system, and the intestine. The inner circle in the center represents the hematopoietic function and immune function within the bone. Blue arrows indicate cell differentiation trajectories, red arrows denote cell migration, and green arrows represent the regulatory effects of one cell type on others. (ER, estrogen; SCFAs, short-chain fatty acids; HSC, hematopoietic stem cell).

**Figure 2 biomedicines-13-02426-f002:**
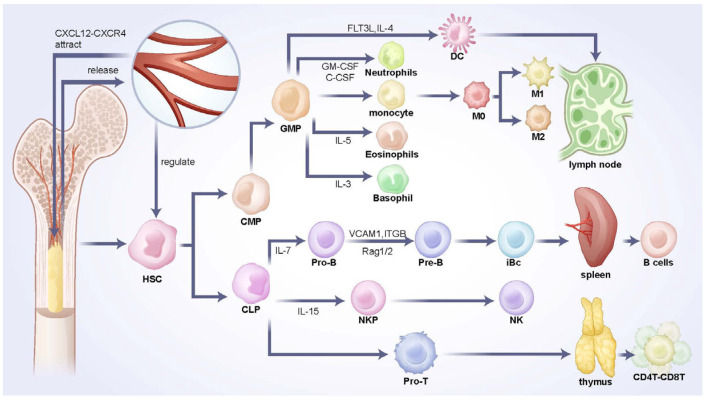
The progress of immune cell development and differentiation in bone marrow cavity. The diagram illustrates the differentiation and maturation trajectory from HSCs to various cell types, including monocyte-macrophages, B cells, and T cells, along with the ultimate destinations of these cells. Key regulatory signaling molecules involved in these processes are also annotated in the figure. (HSC, hematopoietic stem cell; CMP, common myeloid progenitor; CLP, common lymphoid progenitor; GMP, granulocyte-monocyte progenitors; DC, dendritic cell; M0, unpolarized macrophage; M1, M1 macrophage; M2, M2 macrophage; Pro-B, progenitor B cell; Pre-B, precursor B cell; iBc, immature B cell; NKP, natural killer cell progenitor; NK, natural killer cell; Pro-T, progenitor T cell).

**Figure 3 biomedicines-13-02426-f003:**
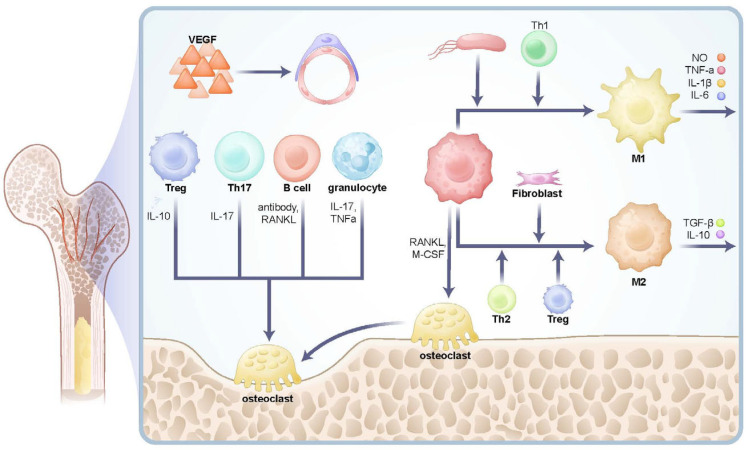
The scheme of immune microenvironment involved in bone regeneration. This figure illustrates the differentiation of macrophages into osteoclasts, which drive osteolytic effects, and the regulatory roles of immune cells such as T cells and B cells in modulating osteoclast activity. It also highlights the distinctive Type S (secondary ossification) vessels specifically associated with cancellous bone. (T-reg, Regulatory T cell; Th1, T helper 1 cell; Th2, T helper 2 cell; Th17, T helper 17 cell; M1, M1 macrophage; M2, M2 macrophage).

**Figure 4 biomedicines-13-02426-f004:**
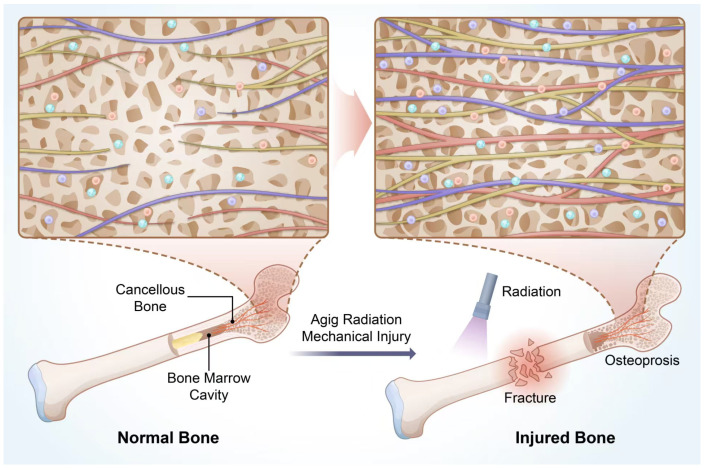
Vessel system regulates the regeneration in bone defect. In cases of injury, such as irradiation, osteoporosis and fractures, the duct system within the cancellous bone will proliferate in large quantities to meet the demands of bone regeneration. During this process, various immune cells will fill the Spaces of the cancellous bone. Cells in different color represent innate immune cells (in purple), MSCs (in red) and lymphocytes (in green).

**Figure 5 biomedicines-13-02426-f005:**
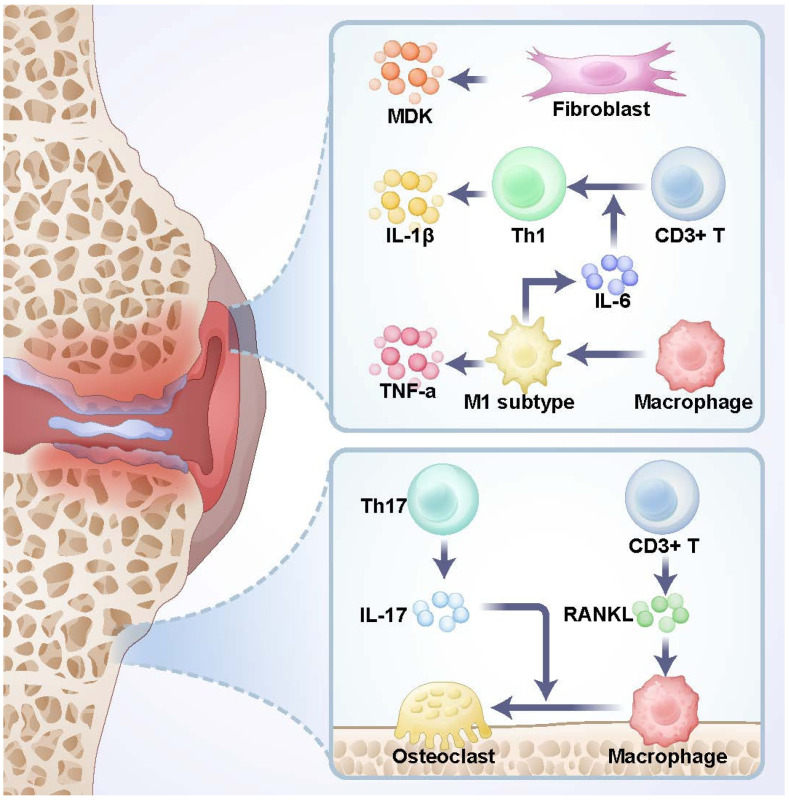
Scheme of the pathogenesis of osteoarthritis involved subchondral bone and synovial. The synovial process mainly involves mononuclear-macrophage infiltration, activation and secretion of inflammatory factors. The subchondral bone process is characterized by macrophage differentiation and the bone remodeling it mediates. (Th1, T helper 1 cell; Th17, T helper 17 cell; MDK, Midkine).

**Table 1 biomedicines-13-02426-t001:** Some other factors contribute to the regulation to bone immunity.

Source	Molecular Classification	Molecular	Target	Function	References
The traumatized brain	Exosomes	miR-21-5p, miR-328a-3p, miR-150-5p	SMAD7, FOXO4, CBL	accelerating bone regeneration	[49,50]
Sympathetic nervous system	Neurotransmitters	Norepinephrine	β2-AR, β1-AR, β3-AR	Enhances RANKL-induced osteoclastogenesis and bone resorption;	[54,55,56,57,58]
Parasympathetic nervous system	Neurotransmitters	Acetylcholine	α7nAChR	Suppresses M1 macrophage polarization; reduces IL-17 secretion by Th17	[65,66,67]
Sensory nerves	Neuropeptide	Substance P	Bone and cartilage cells	Regulates bone metabolism and fracture healing via autocrine/paracrine	[70,71]
Sympathetic nerves	Neuropeptide	NPY	Y1, Y2, Y5 receptors	Inhibits immune cell activation; reduces bone resorption	[73,74]
Central nervous system	Endogenous opioids	Enkephalins	μ-opioid receptors	Modulates sympathetic, parasympathetic activity	[75]
Gut microbiota	Metabolites (SCFAs)	Butyrate, Propionate	HDACs	Inhibit Th17-derived IL-17; promote Treg proliferation and activation.	[82,83,84,85]
Gut microbiota	Tryptophan metabolites	Tryptophan metabolites	aryl hydrocarbon receptor	Promote Treg differentiation; Th17 cell activity; drive M2 and suppress macrophage polarization	[86,87]
Gut microbiota	Polyamines	Spermidine	Foxp3 expression	promote Treg proliferation and differentiation; reduce the secretion of IL-17	[88,89]
Gut MALT	Mucosal immune system	sIgA	Pro-inflammatory factors	Downregulates RANKL and suppresses osteoclast activation	[90]
Ovaries	Hormone	Estrogen	ERα, ERβ receptors	Inhibits osteoclast activity; promotes Treg proliferation	[93]
Thyroid gland	Hormone	T3/T4	Thyroid hormone receptors	Promotes osteoclast formation; regulates Th1 activity	[96,97]
Parathyroid gland	Hormone	Parathyroid hormone	PTH1R	Enhances Th1/Th17 proliferation; increases RANKL expression	[98,99]
Adipocytes	Adipokine	Leptin	Ob-R receptor	Stimulates Th1/Th17 differentiation; enhances osteoclast activity	[110,111,112]
Adrenal glands	Hormone	Cortisol	Glucocorticoid receptor	Suppresses osteoblast function; enhances osteoclast activity and bone loss	[102,103,104]

## Data Availability

Not applicable.

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
