# Peer review of "New Insight into Bone Immunity in Marrow Cavity and Cancellous Bone Microenvironments and Their Regulation"

_biomedicines, 2025, doi:10.3390/biomedicines13102426_

Round 1
Reviewer 1 Report
Comments and Suggestions for Authors
The present study delves into a critical aspect of immune regulation in the maintenance of bone homeostasis. Nonetheless, there are certain deficiencies that preclude the acceptance of MS in its current state. Upon perusal, it becomes evident that the authors display a somewhat superficial comprehension of the subject matter. The lack of specificity in their exposition regarding the impact of various physiological systems, such as the nervous, digestive, and endocrine, on the immune function of bone tissue is particularly noteworthy. In reality, there is no such thing as a singular immune system for bone tissue. Rather, there are pathological processes that involve the immune system operating at various levels, resulting in disruptions in the processes of bone formation and resorption.Moreover, the authors failed to provide a comprehensive analysis of the disparity in the immunological profile of distinct regions of the bone tissue. Why didn't the authors delve into the causes and pathogenesis of rheumatoid arthritis, a bone disease intimately linked to immune system disorders? The captions accompanying the illustrations are woefully inadequate, failing to provide any meaningful explanation of the depicted processes. Moreover, it appears that the illustrations have been derived from the work of other authors, other authors, in which case appropriate references are needed. The text provided does not fully align with the title. There is a lack of clarity in the distinction between Bone immunity within the marrow cavity and the trabecular microenvironments, and there appears to be no originality or novelty in the content.
A few further observations.
- p 3. l.73 Correct the scheme 1 to Figure 1, decipher the abbreviations, and provide a description of the processes shown.
- Please correct the numbering of the figures.
- p 5. l. 157 decipher the abbreviations, and provide a description of the processes shown.
- p 7. l. 221 decipher the abbreviations, and provide a description of the processes shown.
- Table 1 is poorly presented, it is difficult to separate the rows, which makes it very difficult to understand the data.
- The section «The Tubular System and Bone Immunity» presents two opposing viewpoints regarding the presence of lymphatic vessels, which necessitates a re-evaluation in the form of an exploration of diverse perspectives and potential avenues for further investigation into this issue. The paragraph on adipose tissue falls short of being informative; if the authors deem it essential to provide information on how adipose tissue affects the immune system in bone tissue, it is imperative to locate data specifically pertaining to the impact of adipose tissue on immune cells within bone.
- p 13. l. 432 provide a description of the processes shown.
- It is not entirely clear from Sections 2.1 and 2.2 as to why the processes under discussion are compartmentalized in this manner. Is there not a degree of parallelism at work across all domains? The exposition appears somewhat cursory. A more in-depth and comprehensive explication of the phenomena and processes is required. There are many common words, for example, immune cells, bone metabolism, etc.
- In section 1.1. Neuro-Immune-Endocrine Regulation and Mechanisms, well-known regulatory mechanisms of immune cell activity (innate and adaptive mechanisms) are described. However, the detailed action of each mechanism is not provided, and some conclusions, such as the gut having an immunoendocrine function and secreting factors like leptin and IGF-1 to regulate bone immunity, are speculative due to the pleotropic nature of most of these biologically active molecules.
- p 16. l. 558 decipher the abbreviations, and provide a description of the processes shown. Links to the source of the images are required.
Author Response
|
Comments 1: The present study delves into a critical aspect of immune regulation in the maintenance of bone homeostasis. Nonetheless, there are certain deficiencies that preclude the acceptance of MS in its current state. Upon perusal, it becomes evident that the authors display a somewhat superficial comprehension of the subject matter. The lack of specificity in their exposition regarding the impact of various physiological systems, such as the nervous, digestive, and endocrine, on the immune function of bone tissue is particularly noteworthy. In reality, there is no such thing as a singular immune system for bone tissue. Rather, there are pathological processes that involve the immune system operating at various levels, resulting in disruptions in the processes of bone formation and resorption. Moreover, the authors failed to provide a comprehensive analysis of the disparity in the immunological profile of distinct regions of the bone tissue. Why didn't the authors delve into the causes and pathogenesis of rheumatoid arthritis, a bone disease intimately linked to immune system disorders? The captions accompanying the illustrations are woefully inadequate, failing to provide any meaningful explanation of the depicted processes. Moreover, it appears that the illustrations have been derived from the work of other authors, other authors, in which case appropriate references are needed. The text provided does not fully align with the title. There is a lack of clarity in the distinction between Bone immunity within the marrow cavity and the trabecular microenvironments, and there appears to be no originality or novelty in the content. |
|
Response 1: We sincerely thank the reviewer for their valuable comments and constructive suggestions, which have been instrumental in improving our manuscript. We have carefully addressed all the concerns raised and made comprehensive revisions to the text. Key changes and additions are highlighted in red for ease of reference. Below, we provide a point-by-point response to the specific comments: 1.Comment: "The lack of specificity in their exposition regarding the impact of various physiological systems, such as the nervous, digestive, and endocrine, on the immune function of bone tissue is particularly noteworthy." Response: We thank the reviewer for this important observation. To enhance the specificity of our exposition, we have significantly expanded Section 3.1, which details the influence of the nervous, digestive, and endocrine systems on bone immune function. Furthermore, the accompanying table has been updated to provide a clearer summary of the regulatory molecules and their functions originating from different organs. These additions are located on page 8, line 295; page 8, lines 310-326; page 9, lines 331-357; and page 9, line 367.
2.Comment: "The authors failed to provide a comprehensive analysis of the disparity in the immunological profile of distinct regions of the bone tissue. There is a lack of clarity in the distinction between Bone immunity within the marrow cavity and the trabecular microenvironments, and there appears to be no originality or novelty in the content." Response: We appreciate the reviewer's feedback regarding the distinction between bone marrow compartments. First, we have replaced the term "trabecular bone" with "cancellous bone" throughout the manuscript for greater anatomical precision and to avoid potential ambiguity; we apologize for any lack of clarity in our initial terminology. More importantly, to directly address the concern about regional immunological disparity, we have added a new subsection, 2.3 Differences in the Spatial Microenvironment between Cancellous Bone and Bone Marrow Cavity (page 6, lines 245-277). This section provides a comparative analysis of the immune microenvironments in these distinct regions, underscoring the functional specialization and spatial heterogeneity, which we believe enhances the novelty and clarity of our discussion.
3.Comment: "Why didn't the authors delve into the causes and pathogenesis of rheumatoid arthritis, a bone disease intimately linked to immune system disorders?" Response: This is a valid point. While rheumatoid arthritis (RA) is indeed an immune-related disorder with significant bone implications, our review primarily focuses on the bone-centric immune environment. RA is a systemic autoimmune disease whose pathology is predominantly characterized by synovial inflammation and joint destruction. Considering the scope and page limitations of our article, we chose to maintain a stronger focus on the bone marrow and cancellous bone niches. Nevertheless, we have briefly mentioned RA in the discussion section (page 15, line 609) to acknowledge its relevance within the broader field of osteoimmunology.
4.Comment: "The captions accompanying the illustrations are woefully inadequate, failing to provide any meaningful explanation of the depicted processes. Moreover, it appears that the illustrations have been derived from the work of other authors, in which case appropriate references are needed." Response: We thank the reviewer for this critical feedback. We have thoroughly revised all figure captions to provide more detailed and explanatory descriptions of the illustrated biological processes (page 3, line 76; page 5, line 164; page 7, line 238; page 14, line 543; page 17, line 673). Regarding the origin of the illustrations, we would like to clarify that all schematic diagrams included in the review were independently created by synthesizing concepts from the existing literature and integrating our own analytical perspectives. They are original compositions and not direct adaptations from other sources. Therefore, external image references are not required. All specific concepts and findings depicted in the figures are appropriately cited within the main text.
5.Comment: "The text provided does not fully align with the title." Response: We appreciate the reviewer's feedback. In response, we have made extensive revisions throughout the manuscript and have also changed the title to: "New Insights into the Bone Immunity in Marrow Cavity and Cancellous Bone Microenvironments and Their Regulation". We believe these changes ensure a much stronger alignment between the title and the content of the article.
We are deeply grateful for the reviewer's meticulous review and insightful suggestions. We hope that our responses and the revisions made have successfully addressed all concerns and significantly improved the overall quality, accuracy, and comprehensiveness of the manuscript. |
|
Comments 2: p 3. l.73 Correct the scheme 1 to Figure 1, decipher the abbreviations, and provide a description of the processes shown. |
|
Response 2: We thank the reviewer for this valuable suggestion. We have renamed "Scheme 1" to "Figure 1" as recommended. Additionally, all abbreviations used in the figure have been deciphered in the accompanying legend. Furthermore, we have provided a more detailed description of the key biological processes illustrated in the figure. These changes can be found on page 3, line 76 of the revised manuscript.
Comments 3: Please correct the numbering of the figures. Response 3: Thank you for your comments. And we reordered all the Figure and matched the corresponding content.
Comments 4: p 5. l. 157 decipher the abbreviations, and provide a description of the processes shown. Response 4: We thank the reviewer for this suggestion. In response, we have updated the legend for Figure 2 to include a full decipherment of all abbreviations and have added a detailed explanation of the cellular processes shown. We believe these enhancements improve the clarity and informational value of the figure. Please see the updated legend on page 5, line 165-172.
Comments 5: p 7. l. 221 decipher the abbreviations, and provide a description of the processes shown. Response 5: We agree with the reviewer's suggestion. We have deciphered all abbreviations in the figure legend and provided a comprehensive description of the key biological processes illustrated in Figure 3. These revisions have been incorporated on page 7, line 238 of the revised manuscript.
Comments 6: Table 1 is poorly presented, it is difficult to separate the rows, which makes it very difficult to understand the data. Response 6: We agree with the reviewer's assessment regarding the presentation of Table 1. Thank you for this critical feedback. To significantly improve the clarity and readability, we have thoroughly reorganized the table. Specific enhancements include: adjusting row spacing for better visual separation, employing a more legible font, and refining the content for conciseness. The updated table is now presented on page 12.
Comments 7: The section «The Tubular System and Bone Immunity» presents two opposing viewpoints regarding the presence of lymphatic vessels, which necessitates a re-evaluation in the form of an exploration of diverse perspectives and potential avenues for further investigation into this issue. The paragraph on adipose tissue falls short of being informative; if the authors deem it essential to provide information on how adipose tissue affects the immune system in bone tissue, it is imperative to locate data specifically pertaining to the impact of adipose tissue on immune cells within bone. Response 7: Agree. Thank you for your comments. Although the existence of lymphatic ducts has been confirmed by literature, there are still few reports on this study. The research on lymphatic duct reflux for fracture repair is also achieved by ligation of inguinal lymph nodes. It is relatively difficult to directly explore the impact of lymphatic ducts in the cancellous bone on osteogenesis. Therefore, this article mainly refers to these articles as the main sources of viewpoints [1, 2]. In addition, we have supplemented the interaction between intramedullary fat and the immune system. Although there are not many related studies, they all have a significant promoting effect on the research of bone immunity. Such changes were shown at page 11, Line 470-480.
Comments 8: p 13. l. 432 provide a description of the processes shown. Response 8: Agree. Thank you for your comments. And we decipher the abbreviations, and provide a description of Figure 4. Changes were shown at page 14, line 543-546.
Comments 9: It is not entirely clear from Sections 2.1 and 2.2 as to why the processes under discussion are compartmentalized in this manner. Is there not a degree of parallelism at work across all domains? The exposition appears somewhat cursory. A more in-depth and comprehensive explication of the phenomena and processes is required. There are many common words, for example, immune cells, bone metabolism, etc. Response 9: We thank the reviewer for this insightful comment regarding the compartmentalization discussed in Sections 2.1 and 2.2. we have replaced the term "trabecular bone" with "cancellous bone" throughout the manuscript for greater anatomical precision. The bone marrow cavity and cancellous bone share similarities in histology, embryology and morphology. However, bone marrow cavity is relatively immune-exempt, and researchers have not found the presence of lymphoid tissue in the bone marrow cavity. This is the most significant difference between the two. Centering on this point, we made a comparison by comparing the differences in immune function between these two sites. we have added a new subsection, 2.3 Differences in the Spatial Microenvironment between Cancellous Bone and Bone Marrow Cavity (page 6, lines 245-277). Therefore, this is also the key point we want to discuss.
Comments 10: In section 1.1. Neuro-Immune-Endocrine Regulation and Mechanisms, well-known regulatory mechanisms of immune cell activity (innate and adaptive mechanisms) are described. However, the detailed action of each mechanism is not provided, and some conclusions, such as the gut having an immunoendocrine function and secreting factors like leptin and IGF-1 to regulate bone immunity, are speculative due to the pleotropic nature of most of these biologically active molecules. Response 10: Agree. Thank you for your comments. We followed your valuable advice and enriched the content in section you mentioned. Changes were shown at page 8, line 295; page 8, lines 310-326; page 9, lines 331-357; and page 9, line 367. Regarding the statement about the immunoendocrine function of the gut and its secretion of factors such as leptin and IGF-1 to regulate bone immunity, we have strengthened the supporting evidence by citing relevant literature to enhance the credibility of the IGF-1–related conclusion. In consideration of the relatively weaker evidence for leptin, we have removed the corresponding description (page 10, line 422). Additionally, we have explicitly emphasized that these conclusions warrant further experimental validation (page 11, line 424).
Comments 11: p 16. l. 558 decipher the abbreviations, and provide a description of the processes shown. Links to the source of the images are required. Response 11: Agree. Thank you for your comments. And we decipher the abbreviations, and provide a description of Figure 5. Changes were shown at page 17, line 673. The figure in the review are independently drawn by synthesizing insights from previous literature and integrating our own analytical perspectives, rather than directly adapting work from other authors. Therefore, no references for external images are required, and the specific content involved in the figures has been properly cited in the main text. |
|
|
|
4. Response to Comments on the Quality of English Language |
|
Point 1: The English is fine and does not require any improvement. |
|
Response 1: Thank you for your positive feedback on the English language. Prior to submission, we had already engaged experts to polish the article in academic English to enhance its linguistic quality, and we have further refined it based on the review process. |
|
5. Additional clarifications |
|
Thanks for your expert comments. We are so glad to receive review from such a specialist from Biomedicines. We are also thankful for editor’s patient review. We have our article well revised, and we hope it will meet your advice.
|
Refference
- Zheng, Y., et al., A novel therapy for fracture healing by increasing lymphatic drainage. J Orthop Translat, 2024. 45: p. 66-74.
- Zheng Y, et al., Lymphatic platelet thrombosis limits bone repair by precluding lymphatic transporting DAMPs. Nat Commun. 2025 Jan 18;16(1):829. doi: 10.1038/s41467-025-56147-8.

Reviewer 2 Report
Comments and Suggestions for Authors
The review article refers to bone immunity. The article is a basic overview of immune cells and processes involved in bone metabolism, without discussing important pathological events. How does immune cell activation and inflammation affect Calcium absorption and bone density and stability? How do lymphocytes and innate immune cells affect bone tumors? What are the critical issues in osteomyelitis, osteopathy and osteonecrosis involved, and what elements in the trabecular bone can be important to analyze? There are more hormones involved in bone in bone metabolism, a critical one missing is testosterone, other calcitonin, calcitrol, growth hormone, GLP-1 are not even mentioned.
Some sentences should be modified for clarity.
Author Response
|
Comments 1: The review article refers to bone immunity. The article is a basic overview of immune cells and processes involved in bone metabolism, without discussing important pathological events. How does immune cell activation and inflammation affect Calcium absorption and bone density and stability? How do lymphocytes and innate immune cells affect bone tumors? What are the critical issues in osteomyelitis, osteopathy and osteonecrosis involved, and what elements in the trabecular bone can be important to analyze? There are more hormones involved in bone in bone metabolism, a critical one missing is testosterone, other calcitonin, calcitrol, growth hormone, GLP-1 are not even mentioned. |
|
Response 1: We sincerely thank the reviewer for their insightful comments and valuable suggestions, which have greatly helped us improve the depth and quality of our manuscript. We have carefully addressed each point raised, with major revisions highlighted in red throughout the text. Our point-by-point responses are provided below. 1. Comment: How does immune cell activation and inflammation affect Calcium absorption and bone density and stability? Response: We thank the reviewer for this important question. In response, we have expanded our discussion on how immune cells influence osteoclast activation and related mechanisms affecting bone resorption and bone integrity. These additions can be found on page 8, line 295; page 8, lines 310–326; page 9, lines 331–357; and page 9, line 367.
2. Comment 2: The article is a basic overview of immune cells and processes involved in bone metabolism, without discussing important pathological events. How do lymphocytes and innate immune cells affect bone tumors? What are the critical issues in osteomyelitis, osteopathy and osteonecrosis involved, and what elements in the trabecular bone can be important to analyze? Response: We appreciate the reviewer's feedback regarding the scope of pathological discussion. While our review is primarily focused on recent advances in the cellular and molecular mechanisms of bone immunity, we have included discussions of several representative diseases closely related to bone immune dysregulation, such as osteoporosis, osteoarthritis (OA), and hematological malignancies. In order to incorporate cutting-edge insights, we have added a new section, 2.3 Differences in the Spatial Microenvironment between Cancellous Bone and Bone Marrow Cavity (page 6, lines 245–277), which summarizes recent single-cell omics findings on the bone immune microenvironment. Due to space limitations, it was not feasible to exhaustively cover all bone-related pathologies, but we have strengthened the clinical relevance of the immune mechanisms discussed. We would greatly appreciate your understanding in this consideration.
3.Comment: There are more hormones involved in bone metabolism, a critical one missing is testosterone, other calcitonin, calcitriol, growth hormone, GLP-1 are not even mentioned. Response: We agree with the reviewer that these hormones play significant roles and apologize for their omission in the initial version due to space constraints. Following the reviewer's helpful suggestion, we have now included a discussion of the roles of testosterone, growth hormone, calcitonin, and other key hormones in the section on endocrine regulation (page 11, line 435; page 11, line 457) to present a more complete picture of the hormonal regulation network in bone metabolism.
We are truly grateful for the reviewer's thorough and constructive critique. We believe the revisions have significantly enhanced the manuscript's comprehensiveness and scholarly rigor, and we hope the responses satisfactorily address all points raised.
|
|
Comments 2: Some sentences should be modified for clarity. |
|
Response 2: We sincerely thank the reviewer for this valuable feedback. We have carefully revised the manuscript to improve sentence clarity and logical flow throughout, with particular attention to sections where expression may have been ambiguous or complex. For instance, we have replaced the term "trabecular bone" with "cancellous bone" throughout the manuscript for greater anatomical precision and to avoid potential ambiguity. These linguistic and structural enhancements have been systematically applied across all relevant sections of the article to ensure a more coherent, precise, and accessible presentation of our scientific content.
|
|
4. Response to Comments on the Quality of English Language |
|
Point 1: The English is fine and does not require any improvement. |
|
Response 1: Thank you for your positive feedback on the English language. Prior to submission, we had already engaged experts to polish the article in academic English to enhance its linguistic quality, and we have further refined it based on the review process. |
|
5. Additional clarifications |
|
Thanks for your expert comments. We are so glad to receive review from such a specialist from Biomedicines. We are also thankful for editor’s patient review. We have our article well revised, and we hope it will meet your advice.
|
Refference
- Yu S, Yao X. Advances on immunotherapy for osteosarcoma. Mol Cancer. 2024 Sep 9;23(1):192. doi: 10.1186/s12943-024-02105-9.
- Clézardin P, Coleman R, Puppo M, Ottewell P, Bonnelye E, Paycha F, Confavreux CB, Holen I. Bone metastasis: mechanisms, therapies, and biomarkers. Physiol Rev. 2021 Jul 1;101(3):797-855. doi: 10.1152/physrev.00012.2019.

Round 2
Reviewer 1 Report
Comments and Suggestions for Authors
The authors have made significant improvements to the manuscript. While it is acknowledged that over time this review will need to be further completed, expanded, and refined, the article is currently suitable for publication.
Reviewer 2 Report
Comments and Suggestions for Authors
The authors have responded to the queries and modified the article accordingly